# One- and Two-Month-Old Dog Puppies Exhibit Behavioural Synchronization with Humans Independently of Familiarity

**DOI:** 10.3390/ani12233356

**Published:** 2022-11-29

**Authors:** Charlotte Duranton, Cécile Courby-Betremieux, Florence Gaunet

**Affiliations:** 1Centre de Recherche Textes et Cultures, Université d’Artois, 62030 Artois, France; 2Centre D’élevage et D’éducation Jacques Bouniol, Ecole des Chiens Guides de Paris, 78530 Buc, France; 3Laboratoire de Psychologie Cognitive, CNRS, Université Aix-Marseille, Federation 3C, 13331 Marseille, France

**Keywords:** dog–human synchronization, interspecific synchronization, domestic dogs, behavioural mimicry, affiliation, ontogeny, development

## Abstract

**Simple Summary:**

Behavioural synchronization is a widespread skill in social species as it helps increase group cohesion. It is highly studied among humans, but should be investigated from an interspecific perspective to better understand the influence of both phylogeny and ontogeny in the phenomenon. In humans, behavioural synchronization appears from a young age between babies and their mothers, and is essential for the infants to learn social skills. Whereas it has been recently evidenced that dogs synchronize their behaviour with humans, little is known about the presence of such a phenomenon from a young age. We thus investigated the existence of behavioural synchronization between puppies and humans. We did find that puppies of one and two months of age synchronize their locomotor activity with that of humans. Such findings have significant theoretical and practical applications, and can be used from a societal perspective to increase positive cohabitation between both species.

**Abstract:**

Behavioural synchronization is a widespread skill in social species as it helps increase group cohesion among individuals. Such a phenomenon is involved in social interactions between conspecifics as well as between individuals from different species. Most importantly, familiarity and affiliation between interacting partners influence the degree of behavioural synchronization they would exhibit with each other. For example, in human–dog dyads, the more a dog is affiliated with its human partner, the more it behaves in a synchronous way with them. However, little is known about the ontogeny of such a behaviour, especially from an interspecific perspective. The aim of the present study was thus to investigate the existence and modalities of activity synchrony, a type of behavioural synchronization, between humans and puppies. To do so, we observed 29 dog puppies interacting with two different humans (familiar and unfamiliar experimenters). Puppy movements and general activity in relation to the human ones were observed. Results evidenced that puppies did exhibit locomotor synchrony with humans, but familiarity did not affect its degree. It is the first time that activity synchrony with human walk is evidenced in puppies, highly suggesting that dogs’ ability to behave in synchronization with humans seems to be genetically selected through the process of domestication, while the effect of familiarity on it might develop later during the individual ontogeny.

## 1. Introduction

Behavioural synchronization is a widespread phenomenon which broadly consists of doing the same thing (activity synchrony), in the same place (local synchrony) and at the same time (temporal synchrony) as others [1]. Such a skill is highly adaptive as it increases survival chances of individuals by decreasing predation pressure as well as increasing social group cohesion [1,2,3]. In humans, the more people are affiliated, the more they synchronize their behaviour, and reciprocally, synchronized behaviour indicates a high level of affiliation [4,5,6,7,8]. The importance of affiliation is highlighted in mother–infant interactions, and plays an important role in communication. For example, human babies synchronize their movements with those of their mother when she speaks to them; by doing so, they prepare their perception–action system for language acquisition [9]. Behavioural synchronization thus acts as a “social glue” [7] and is essential for promoting learning between group members.

Although its importance is acknowledged in ethology when considering conspecifics in humans and various species, little is known about behavioural synchronization at an interspecific level. It is nevertheless essential to better investigate the respective roles of phylogeny and ontogeny in such an important social phenomenon; thus, it has recently started to be studied between dogs and humans. Both species share strong affiliative bonds and dogs have evolved social abilities to communicate with humans, including a unique sensitivity to our behavioural cues [10,11,12]. Behavioural synchronization promotes affiliation between dogs and humans: dogs prefer people who synchronize their activity with them [13]. Note that it is a reciprocal effect, as people prefer dogs who synchronize with them [14]. It has also recently been observed that dogs synchronize their activity with humans in various situations [15]. Both indoors and outdoors, dogs synchronize their gaze and movements with those of humans [15,16,17]. When highly familiar with their human partner, dogs also synchronize with their reaction when facing an unfamiliar object [18] or human [19]. As found between humans, degree of familiarity does play an essential role between dogs and humans. In contrast, dog–human dyads who are less familiar exhibit low degree of synchrony [17,20,21].

It has thus been suggested that dogs possess behavioural synchronization skills similar to those of humans. However, the respective roles of phylogeny and ontogeny in the phenomenon are not known yet [22] and would be important to study. To investigate such a question, we decided to investigate the existence of activity synchrony in puppies of different ages when walking with humans, and its link with familiarity. Same puppies were observed at ages of one and two months. Given that puppies start walking around 21 days of age [23], we expected to observe puppy–human behavioural synchronization from a young age, e.g., one month of age. Considering the effect of life experiences on activity synchrony in adult dogs, we hypothesized that it will impact puppy behavior, too. We expected that familiarity of the experimenter would affect the puppies’ interactions with her. More precisely, we expected that puppies would exhibit more activity synchrony with the familiar experimenter compared to the unfamiliar one, and at a stronger level in two-month-old puppies compared to one-month-old ones.

## 2. Methods

### 2.1. Participants

In total, 29 dog puppies (17 males and 12 females) from 5 litters of retriever dogs were tested from September 2018 to July 2019. Each puppy was tested once at one month of age and once at two months. The experiments took place at the Center for breeding and education Jacques Bouniol, Guide Dogs School of Paris (Buc, France), all in the same testing room (3.95 m × 2.40 m). Puppies were usually housed with their mother and siblings, in the testing room with an outdoor access which was sometimes opened in summer. To ensure puppies were all tested in the same conditions, the outdoor access was closed during the short testing periods. Each puppy was regularly controlled for any health issue by the breeding facility’s caregiver as well as by an external veterinary. If a puppy was feeling unwell or stressful/fearful, it was not tested, but it did not happen during the testing time.

In order to test the effect of familiarity on the level of synchronization, Cécile Courby-Betremieux (C.C.), the caregiver of the litters, was the familiar experimenter (she interacted with all puppies on a daily basis, providing them food, medical care, as well as playing and hugging sessions), and Charlotte Duranton (C.D.) was the unfamiliar one during the tests.

### 2.2. Ethical Note

The present study was only observational and puppies were not subjected to any physical or psychological harm in the course of the observations. The study was conducted in accordance with the legal requirements of France for animal welfare (Rural Code Article R214-17, and of the official French Legal Code of Animals, 2018) and the institutional guidelines of the Aix-Marseille Université, France, as well as of the guide Dogs School of Paris, France. Each puppy was free to move throughout the testing area (their familiar playing room) without any physical constraints. We visually controlled for stress-associated behaviors as puppies were tested in situations as similar as possible with their usual daily routine. No physical manipulation nor sampling was performed on any of the puppies participating in the study (e.g., blood or saliva sampling).

### 2.3. Set up and Procedure

C.D. explained the precise procedure to C.C. in the absence of the puppies. The room was familiar for the puppies, but was emptied for the experimental purpose (see Figure 1). Thus, each puppy had 15 min to explore the room freely in order to become familiar with its emptiness, divided as follows: 13 min all together with their mother to avoid stress, then 2 min alone.

There were two different conditions, each subdivided into two parts in order to ensure that puppies could be able to maintain their attention according to their cognitive abilities. We set them to a duration lasting 20 s; indeed, exercises to test the ability to maintain attention were found to be 30 s long in adult dogs [24]:Human Still Condition: The human experimenter was staying still without interacting with the puppy (40 s).-Standing Still Part: the experimenter was instructed to stand still at a predefined location for 20 s (see Figure 2A).-Sitting Still Part: the experimenter was instructed to sit on a chair at a predefined location for 20 s (see Figure 2A).Human Move Condition: The human experimenter was walking without interacting with the puppy (40 s).-Normal Walk Part: the experimenter was instructed to walk for 20 s according to a predefined path at a normal walking speed (see Figure 2B).-Fast Walk Part: the experimenter was instructed to walk for 20 s at a fast speed (but without running) according to the same predefined path (see Figure 2B).

The data from Standing Still and Sitting Still parts on one hand and Normal Walk and Fast Walk parts on the other hand were thus pooled for Human Still and Human Move conditions, respectively, for data analyses.

The order of all conditions was randomly assigned to each dyad. The experimenters (C.C. and C.D.) were informed when they had to switch to the next condition due to a phone connected by earphones, and the application “Seconds” which was previously programmed to ring every 20 s. During the entire test (80 s), the puppies were free to move without physical constraint. The experimenter was instructed to remain neutral without showing any emotional reactions and to neither talk to the puppies nor look at them, not to influence their behaviour. All puppies were tested only once with C.C. and once with C.D. (order counterbalanced across puppies). All trials were video recorded using an Iphone 10^®^, placed with an angle allowing to see the testing area and the puppy, but not showing the faces of the experimenters in order to ensure an impartial video coding (see Figure 2).

### 2.4. Behavioural and Statistical Analysis

All videos were analysed by using Solomon Coder software (copyright by András Péter, http://solomoncoder.com/).

Studied variables were: i. Close: Time spent by the puppy less than 1 m around the experimenter (whatever the position or activity); ii. Moving: Time spent moving by the puppy (whatever its exact gait: walking, trotting or running); iii. Moving fast: Time spent moving fast by the puppy (only trotting or running).

Walking is defined as a four-beat gait in which each paw hits the ground independently, e.g., left anterior, right posterior, right anterior, left posterior. Trotting is considered a faster gait than walking and defined as a two-beat gait, e.g., right anterior and left posterior simultaneously, then projection phase, then left anterior and right posterior simultaneously. Running is considered a faster gait than walking and trotting and defined as a three-beat gait, e.g., left posterior, then left anterior and right posterior simultaneously, then right anterior, then projection phase.

Firstly, we analysed behavioural synchronization at a general level, comparing the puppies’ general movement (Close and Moving) in the two experimenting conditions: Human Still and Human Move. Secondly, we analysed a more precise activity synchrony by comparing the time spent moving fast by the puppies (Moving fast) according to the experimenter’s different speed of walk (Normal Walk and Fast Walk).

Statistical analyses were made by using R software (version 3.2.0, The R Foundation for Statistical Computing, Vienna, Austria, http://www.r-project.org, accessed on 7 September 2022). As variables were not normally distributed and data included individuals as a dependent factor, we used LMEs with an Anova type 3 to test the effects of condition and its potential interactions with age or familiarity. As focusing on the existence of behavioural synchronization and its functional properties, we focused the statistical analyses only on the effect of condition, and any possible interaction with it: condition, condition*familiarity, condition*sex, condition*age. We used backward elimination to remove non-significant interactions from the models. When there was a significant effect of condition, we carried out post hoc comparisons with Holm–Bonferroni corrections for multiple tests.

## 3. Results

### 3.1. General Behavioural Synchronization

#### 3.1.1. Close

We found no significant interaction between condition and familiarity (LME with permutation tests, *p* = 0.822) as well as no effect of condition alone on time spent by the puppies close to the experimenter (LME with permutation tests, *p* = 0.099). To ensure that puppies were not afraid of the unfamiliar experimenter, we controlled here the effect of familiarity alone: puppies stayed close to both experimenters for the same amount of time (LME with permutation tests, *p* = 0.677). We did not find a significant interaction with sex (LME with permutation tests, *p* = 0.873). However, we found a highly significant effect of age on the time spent by the puppies close to the experimenter: two-month-old puppies spent significantly more time close to the experimenter compared to one-month-old ones (mean_1m_ = 24.78 ± 0.94 s, mean_2m_ = 30.35 ± 2.81 s, LME with permutation tests, *p* = 0.000).

#### 3.1.2. Moving

We found a highly significant interaction between condition and puppies’ age (LME with permutation test, *p* = 0.000). Puppies of one month of age spent significantly more time moving when the experimenter was moving than when she was still (mean_mov1m_ = 19.02 ± 1.12 s, mean_still1m_ = 11.88 ± 0.93 s, LME with permutation test, *p* = 0.000). Same difference was observed when the puppies were two months of age (mean_mov2m_ = 28.67 ± 1.22 s, mean_still2m_ = 12.2 ± 0.82 s, LME with permutation test, *p* = 0.000). We found that whereas the time of the puppies’ movement when getting older did not differ while the experimenter was still (mean_still1m_ = 11.8 ± 0.93 s, mean_still2m_ = 12.2 ± 0.82 s, LME with permutation test, *p* = 0.788), it did highly differ when the experimenter was moving (mean_mov1m_ = 19.02 ± 1.12 s, mean_mov2m_ = 28.67 ± 1.22 s, LME with permutation test, *p* = 0.000), see Figure 3.

We found no significant interaction with familiarity (LME with permutation tests, condition*familiarity *p* = 0.891) and time spent moving fast by the puppies as well as no correlation with puppies’ sex (LME with permutation tests, *p* = 0.733).

## 4. Precise Activity Synchrony

### Moving Fast

We found a highly significant interaction between the condition and puppies’ age (LME with permutation test, *p* = 0.000). One-month-old puppies spent as much time moving fast when the experimenter was exhibiting a fast walk as with a normal walk (mean_nwk1m_ = 0.95 ± 0.12 s, mean_fwk1m_ = 1.6 ± 0.21 s, LME with permutation test, *p* = 0.072), whereas two-month-old puppies exhibit a highly significant difference (mean_nwk2m_ = 0.36 ± 0.04 s, mean_fwk2m_ = 3.13 ± 0.41 s, LME with permutation test, *p* = 0.000). Two-month-old puppies were exhibiting shorter times moving fast than one-month-old ones in the Normal walk part (mean_nwk1m_ = 0.95 ± 0.12 s, mean_nwk2m_ = 0.36 ± 0.04 s, LME with permutation test, *p* = 0.024) whereas they were exhibiting longer moving fast times in the Fast walk part (mean_fwk1m_ = 1.6 ± 0.21 s, mean_fwk2m_ = 3.13 ± 0.41 s, LME with permutation test, *p* = 0.005), see Figure 4.

We found no significant interaction with familiarity (LME with permutation tests, condition*familiarity, *p* = 0.486) on time spent moving fast by the puppies, as well as no interaction with puppies’ sex (LME with permutation tests, *p* = 0.400).

## 5. Discussion

The present study evidenced for the first time that one- and two-month-old puppies do exhibit behavioural synchronization with humans. More precisely, they synchronize their activity with humans by moving significantly longer when the experimenter is moving compared to when she is not. When getting older, they do it precisely, as two-month-old puppies modify their pace of movement: they move slowly when the human walks at a regular pace, and faster when the human is moving fast.

Such findings are consistent with a recent study evidencing social referencing in puppies [25], a general level of behavioural synchronization [19]. Our results are also in line with previous studies evidencing that adult dogs exhibit activity synchrony with humans when they are moving [16,22]. However, it is striking that such a precise activity synchrony is evidenced in puppies. We can thus suggest that, from a phylogenetic perspective, the fact that such young puppies already exhibit behavioural synchronization with humans might be due to genetic selection through domestication. It has indeed been suggested that dogs’ general ability to engage in activity synchrony involving movement, such as following humans when they move from camp to camp, or during hunting sessions, has been selected through their evolution for living together with humans [11,13,26]. It is acknowledged that domestication has affected dogs’ human-directed social behaviors [27,28]; we thus hypothesize that dogs’ behavioural synchronization with humans is also applicable. This would be plausible with one of our previous hypotheses suggesting that behavioural synchronization has been selected early in the domestication history of the species as it is adaptive and found in all dogs [13]. However, the present study only tested one type of puppies selected for working close to humans. To further investigate potential effect of genetic selection, we can suggest further research to test different breeds of puppies, as well as retriever puppies from different breeding lineages (such as hunting, guiding, companioning, etc.). The fact that two-month-old puppies exhibit more subtle activity synchrony with humans compared to one-month-old ones is not in contradiction with such a hypothesis. It is most likely due to a difference in locomotor control at such ages. At one month, puppies do not completely master their coordination: adult-like coordinated walk appears around 40 days of life [23] and motor cortex is mature around three months of life [29]. One-month-old puppies have more difficulty coordinating their paw movements and they might have to engage in fast gaits such as trotting or running to follow a walking human, even at a normal pace, whereas two-month-old puppies are adult-like coordinated, which clearly appears in the fact that they modulate their fast gaits according to the human ones. It is also highly possible that life experiences impact such a behavior: we suggest that two-month-old puppies may have interacted with humans more, and by doing so may have learned that it is positive for them to run when a human moves fast, e.g., in playing contexts, for example. Repetition of such positive situations through time may have reinforced their activity synchrony, likely indicating the presence of a mature resonance motor system as in primates [30,31]. Note that we did not find any effect of puppy sex on their degree of synchronization. It is in line with the fact that sexual maturity is not achieved yet and that hormones have little impact on young dogs’ behaviors [32,33,34]. It is also consistent with the fact that tested puppies do not live as pets, and are thus not influenced by the owners’ different treatments that could modify their behaviour in terms of behavioural synchronization [19].

However, contrary to what we expected and was previously found in adult dogs [15,17] we did not find any effect of familiarity on the degree of activity synchrony exhibited by the puppies towards humans. We indeed observed that puppies exhibit a similar level of activity synchrony with the familiar as well with the unfamiliar human and stayed similarly close to both of them. Such findings suggest that even if the ability to synchronize with humans may be genetically selected, the influence of affiliation on the degree of synchrony appears through individual life experience. In our study, as puppies were from a guide dog’s breeding, even at two months they were not living in human homes: they were staying in their enclosure, interacting with their caregiver from time to time (e.g., when feeding, training, etc.). Such familiarity may not be sufficient to create a strong affiliative/attachment bond that would have impacted the level of behavioral synchronization [35]. We thus recommend that further studies test more precise degrees of human–puppy familiarity, as well as older puppies (three- or four-month-old) living as pets in human families to investigate a potential effect of stronger affiliation compared to the present study.

Finally, it can also be argued that such observed synchrony is explained by proximity seeking: puppies would only move to remain close to the human experimenter. Proximity seeking behaviors are mainly encountered in situations perceived as stressful by the animal [15,36]. It is thus unlikely in the present study as puppies were tested in a very familiar area, in conditions as similar as possible as those of their usual daily routine, and we visually controlled that the puppies did not display stress-related behaviors. Additionally, if proximity seeking was implied, we would have found an effect of condition on the time spent close to the experimenter. Puppies would have spent more time close to the experimenter when she was not moving, especially at one month, as it would have been easier for them to stay in proximity. In addition, if proximity seeking were at play, puppies would have exhibited more time close to the familiar experimenter compared to the unfamiliar one, as it is known that a familiar individual provides more reassurance [35,37,38,39]. We nevertheless encourage further studies to control for stress-associated physiological parameters in order to properly control for proximity seeking.

Overall, the present study evidenced the existence of interspecific behavioural synchronization between humans and dog puppies. Such findings have theoretical significance as they allow a better understanding of the mechanisms underlying behavioural synchronization. Further research is now needed to investigate the effect of proper affiliation (by testing older pet puppies) and to generalize it to the whole species (by testing puppies from different breeds and types). Our findings also have practical applications from a societal perspective, e.g., for dog training practices by encouraging people to create a positive affiliative bond with their puppies, allowing a smoother management of the dogs from a young age, and reducing the risk of problematic behaviors in adult dogs.

## 6. Conclusions

In conclusion, the present study is the first one to evidence activity synchrony with humans in one- and two-month-old dog puppies. However, no effect of familiarity between the interacting partners was found on the degree of the puppies’ behavioural synchronization. We thus suggest that behavioural synchronization is a general social skill selected during domestication which appears at a very young age in dogs, but which is then modulated by life experiences: the effect of affiliation might appear later in dogs’ ontogeny.

## Figures and Tables

**Figure 1 animals-12-03356-f001:**
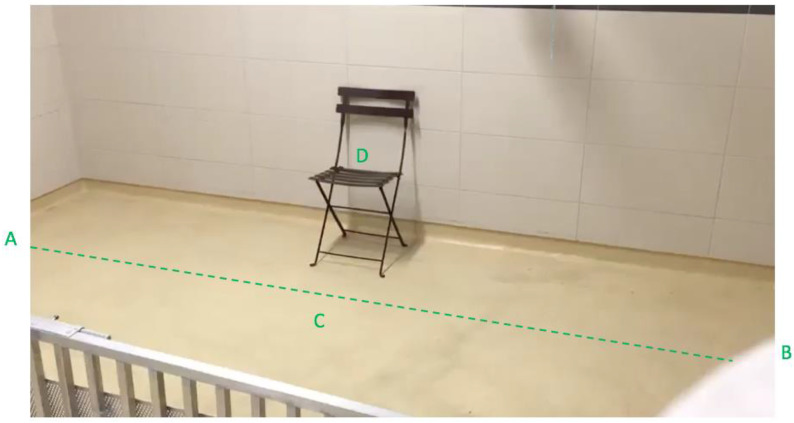
Experimental setup. A and B: Predefined line to be walked by the experimenter during Human Move condition. C and D: Experimenter locations for the Human Still condition (respectively for the Standing Still part and the Sitting Still part).

**Figure 2 animals-12-03356-f002:**
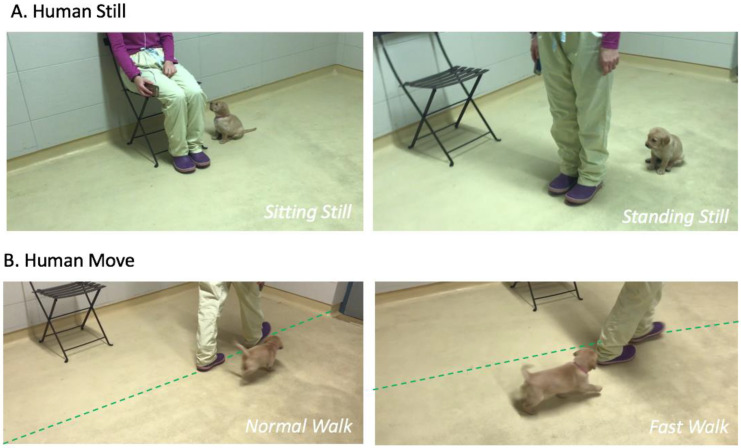
Testing conditions. (**A**): Human Still condition. The experimenter is staying still at the predefined locations. (**B**): Human Move condition. The experimenter is walking along the predefined line (added in green only for the photo).

**Figure 3 animals-12-03356-f003:**
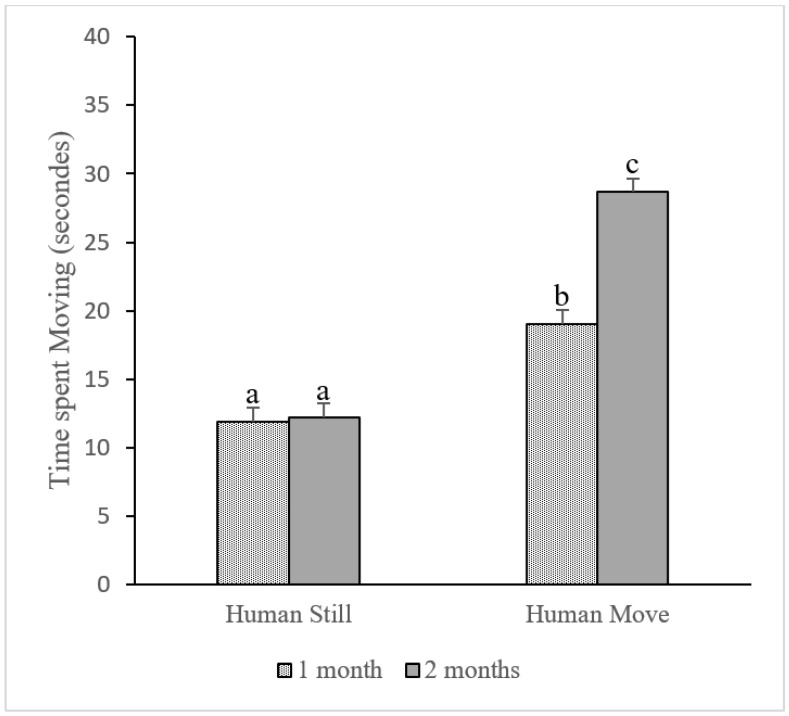
Time spent moving by the puppies. Puppies generally moved significantly more when the experimenter was walking compared to when she was still. The moving behavior of one- and two-month-old puppies did not differ when the experimenter was still (LME with permutation test, *p* > 0.05); however, it did significantly differ when she was moving (LME with permutation test, *p* = 0.000). Graphics represent means + standard errors (seconds). Different letters mean significant differences.

**Figure 4 animals-12-03356-f004:**
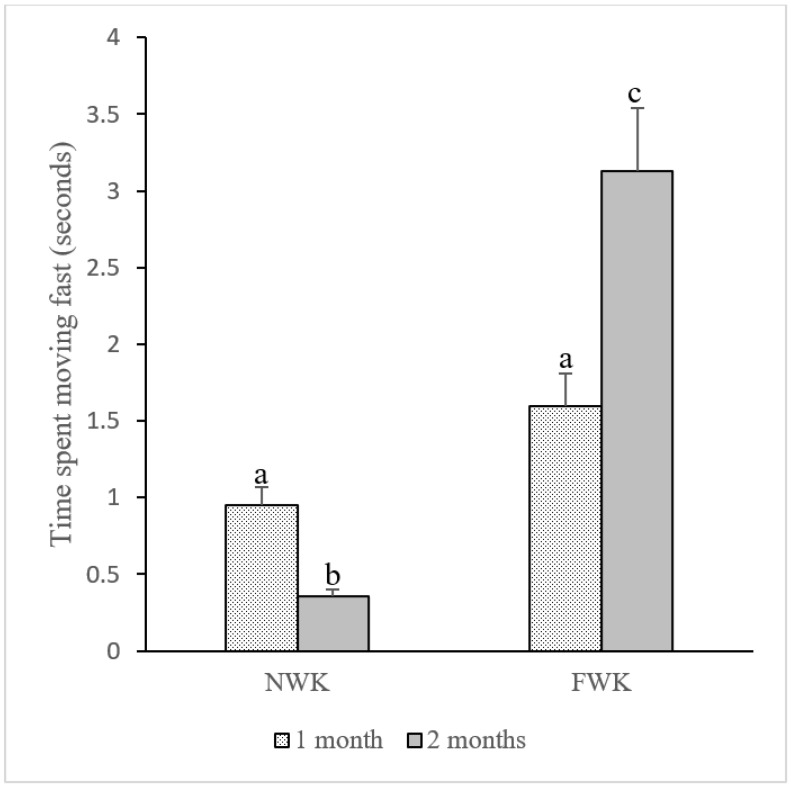
Time spent moving fast by the puppies. Two-month-old puppies exhibited significantly more trotting and running when the experimenter was walking at a fast pace compared to a normal pace (LME with permutation test, *p* = 0.000), whereas one-month-old puppies did not exhibit such a difference (LME with permutation test, *p* = 0.072). NWK: Normal walk, FWK: Fast walk. Graphics represent means + Standard Errors (seconds). Different letters mean significant differences.

## Data Availability

The data presented in this study are available on request from the corresponding author.

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
