# Peer review of "One- and Two-Month-Old Dog Puppies Exhibit Behavioural Synchronization with Humans Independently of Familiarity"

_animals, 2022, doi:10.3390/ani12233356_

Round 1

Reviewer 1 Report

I agree that the lack of effect of familiarity on synchronization may be due to a relatively small difference between the level of past interaction with familiar vs. unfamiliar human. It would be helpful to add some more detail about the level of contact and interaction associated with the role of 'caregiver'. In addition, future research should more definitively differentiate these conditions, e.g. have a category of greater 'bonding' with familar person through handling, cuddling, treating etc.

Reviewer 2 Report

Dear Authors,

Thank you for this very interesting article. Your research explored behavioural synchronisation between 1 and 2 month old puppies and familiar and unfamiliar humans. The paper is nicely written and easy to read and follow, although there are occasional spelling and grammatical issues to correct. The introduction/literature review, methodology, and discussion sections are very good. Your results section could be edited somewhat to improve reproducibility. I feel that I must comment on your ethical note (lines 75-81). Although it is true that your study was observational and may pass the “needle stick” criterion for ethical review, there was potential for emotional distress. Although the puppies were tested in a room familiar to them, isolation from their mother and littermates may have caused stress. Providing data on the behavioural signals observed during each test would help to assuage this concern. On the other hand, I must acknowledge that at 1 and 2 months old, puppies venture further from their mothers and start exploring the world as part of normal socialisation. Exposure to an unfamiliar person, especially one that is moving and perhaps acting strangely, could also cause stress. However, I must also acknowledge that meeting unfamiliar people (who interact appropriately with them) is also part of normal puppy socialisation and could be considered appropriate (so long as the puppy has some control and the ability to withdraw from that person). So, there were some additional factors/potential risks to consider. Why did you not seek approval from an ethical review committee? If you do have approval, please specify this and provide the approving body and approval number in the manuscript. If not, I recommend that you edit your ethical note to address these concerns. When you use the term “affiliation” throughout the article, do you mean attachment? If so, attachment to the caregiver could have been objectively assessed. If you are using the term more generally, I would suggest changing it to “familiar” or “familiarity” or a variation of this as appropriate throughout. I have included some additional specific comments on the manuscript below. I enjoyed your article and hope that my feedback is helpful.

Specific comments:

1.    Lines 67-71 (Methods):

How were the puppies housed, fed, and cared for? Were they examined by a veterinarian prior to participation in your study? Did you have any inclusion or exclusion criteria (e.g. ill health, fearful behaviour)?

2.    Lines 75-81 (Ethical note): See previous comments.

Also, cite and reference the legislation that was adhered to.

3.    Line 111 (Methods): “The order of each condition was randomly assigned to each dyad.”

Was the order of all conditions randomised or just the movement conditions?

4.    Lines 118-119 (Methods): “All trials were video recorded”.

Using what equipment? How many video recorders/cameras? How were they set up in the room?

5.    Line 154 (Results):

For reproducibility/checking of statistics, I recommend including a table featuring the time spent in proximity, time moving, and time moving fast for each puppy.

6.    Lines 163-166 (Results): “However, we found a significant effect of age on the time spent by the puppies close to the experimenter: two-months old puppies spent significantly more time close to the experimenter compared to one-month old ones (mean1m= 24.78 ± 0.94 seconds, mean2m= 30.35 ± 2.81 seconds, LME with permutation tests, P= 0.000).”

P= 0.000 is very highly significant, not just significant

7.    Lines 169-177 (Results): “Puppies of one month-old spent significantly more time moving when the experimenter was moving than when she was still (meanmov1m= 19.02 ± 1.12 seconds, meanstill1m= 11.88 ± 0.93 seconds, LME with permutation test, P= 0.000). Same difference was observed when the puppies were two months-old (meanmov2m= 28.67 ± 1.22 seconds, meanstill2m= 12.2 ± 0.82 seconds, LME with permutation test, P= 0.000) We found that whereas puppies did not differ in their time moving while the experimenter was still when getting older (meanstill1m= 11.8 ± 0.93 seconds, meanstill2m= 12.2 ± 0.82 seconds, LME with permutation test, P= 0.788) they did differ when the experimenter was moving (meanmov1m= 19.02 ± 1.12 seconds, meanmov2m= 28.67 ± 1.22 seconds, LME with permutation test, P= 0.000), see Figure 3.”

P= 0.000 is very highly significant, not just significant

8.    Lines 190-195 (Results):

P= 0.000 is very highly significant, not just significant

9.    Lines 221-223 (Discussion): “We can thus suggest that, from a phylogenetic perspective, the fact that such young puppies already exhibit behavioural synchronization with humans might be due to genetic selection through domestication.”

You could actually test selection somewhat using your current data set. Were there differences between puppies from different litters?

10. Lines 229-231 (Discussion): “This would be plausible with one of our previous hypotheses suggesting that behavioural synchronization has been selected early in the domestication history of the species as it is adaptive and found in all dogs”

This cannot be stated categorically for all dogs, as you only tested this in one breed of dog, and presumably in lines that have been carefully selected for working closely with humans.

11. Lines 233-234 (Discussion): It is most likely due to a difference in locomotor control in such ages.” And lines 255-257: “Such findings suggest that even if the ability to synchronize with humans may be genetically selected, the influence of affiliation on the degree of synchrony appears through individuals’ life experience.”

These statements contradict each other (physical ability versus learned behaviour). Reword these to indicate both are possible contributing factors.

12. Lines 267-268 (Discussion): It is thus unlikely in the present study as puppies were tested in a familiar area, and we visually controlled that the puppies did not display stress-related behaviours.”

Separation from their mothers and litter mates and exposure to unfamiliar humans may cause stress. If you have data on behavioural observations this should be included in the results or in an appendix.

13. Discussion (overall):

·      What is the significance/application of this new knowledge?

·      What are the limitations of your study?
